# Group I mGluRs in Therapy and Diagnosis of Parkinson’s Disease: Focus on mGluR5 Subtype

**DOI:** 10.3390/biomedicines10040864

**Published:** 2022-04-07

**Authors:** Shofiul Azam, Md. Jakaria, JoonSoo Kim, Jaeyong Ahn, In-Su Kim, Dong-Kug Choi

**Affiliations:** 1Department of Applied Life Science, Graduate School, BK21 Program, Konkuk University, Chungju 27478, Korea; shofiul_azam@hotmail.com (S.A.); md.jakaria@florey.edu.au (M.J.); kgfdkr@gmail.com (J.K.); neverland072@kku.ac.kr (J.A.); 2Melbourne Dementia Research Centre, The Florey Institute of Neuroscience and Mental Health, The University of Melbourne, Parkville, VIC 3052, Australia; 3Department of Biotechnology, College of Biomedical and Health Science, Research Institute of Inflammatory Disease (RID), Konkuk University, Chungju 27478, Korea

**Keywords:** glutamate signalling, metabotropic glutamate receptors, C G-protein-coupled receptors, neurodegenerative diseases, positron emission tomography, radioligands

## Abstract

Metabotropic glutamate receptors (mGluRs; members of class C G-protein-coupled receptors) have been shown to modulate excitatory neurotransmission, regulate presynaptic extracellular glutamate levels, and modulate postsynaptic ion channels on dendritic spines. mGluRs were found to activate myriad signalling pathways to regulate synapse formation, long-term potentiation, autophagy, apoptosis, necroptosis, and pro-inflammatory cytokines release. A notorious expression pattern of mGluRs has been evident in several neurodegenerative diseases, including Alzheimer’s disease, Parkinson’s disease, Huntington’s disease, and schizophrenia. Among the several mGluRs, mGluR5 is one of the most investigated types of considered prospective therapeutic targets and potential diagnostic tools in neurodegenerative diseases and neuropsychiatric disorders. Recent research showed mGluR5 radioligands could be a potential tool to assess neurodegenerative disease progression and trace respective drugs’ kinetic properties. This article provides insight into the group I mGluRs, specifically mGluR5, in the progression and possible therapy for PD.

## 1. Introduction

Glutamate, the most important excitatory neurotransmitter of the mammalian central nervous system (CNS), has a critical role in developing memory and synaptic plasticity. However, glutamate hyperactivation could precede and/or exaggerate neurodegenerative disease pathology [1,2]. There are two distinct glutamate receptors, namely ionotropic glutamate receptors (iGluRs) and metabotropic glutamate receptors (mGluRs). Unlike iGluRs, which are ligand-gated ion channels that promote excitatory neurotransmission rapidly [3], mGluRs promote G-protein uncoupling. mGluRs uncouple Gα-βγ G-proteins and increase Gα-mediated intracellular second messenger level or βγ-mediated ion channel regulation and stimulate non-canonical pathways [4,5]. The mGluRs belong to class c G-protein-coupled receptors (GPCRs), and so far, eight subtypes have been identified. These subtypes are further divided into three sub-categories according to phenotypes and intracellular signalling [6,7,8]. Group I consists of mGluR1 and mGluR5 that couple to Gα_q/11_ G-proteins, promoting intracellular Ca^2+^ efflux [9,10]. Group II contains mGluR2 and mGluR3; and mGluR4, mGluR6, mGluR7, and mGluR8 belong to group III mGluRs [8]. Both group II and III mGluRs negatively regulate adenylyl cyclase via Gαi, and they can inhibit glutamate or γ-aminobutyric acid (GABA) release via auto-receptor action [11].

Parkinson’s disease (PD), the second most prevalent neurodegenerative disease, is characterised by motor and non-motor disability manifestation, and this chronic progressive neurodegenerative disease affects mostly older adult people but could also affect younger people. Mounting evidence suggests that glutamate and dopamine regulate neurotransmission in the nigrostriatal, mesocortical, and mesolimbic systems [1,2,3,4]. However, this mutual signalling has been shown to conspicuously affect PD [5], where increased mGluR expression led to the poisoning of dopaminergic neurons in the substantia nigra [6]. Increased glutamate release, at the pathological condition, due to impaired glutamate reuptake at the presynaptic membrane, increases extracellular glutamate concentration. Excessive glutamate release could increase Na^+^ and Ca^2+^ concentration, and that could directly induce neuronal cell death and neurodegeneration in PD. In addition, activated microglia and reactive astrocytes can exacerbate the condition by increasing the large volume of glutamate release.

Considerable evidence indicates that pharmacological inhibition by glutamatergic antagonists or negative allosteric modulation of group 1 mGluRs have been shown to protect dopaminergic neurons and ameliorate dyskinesia in PD animal models [12,13,14]. Specifically targeting mGluR5 could ameliorate motor and/or cognitive impairment. These studies suggest that the anomalies in group 1 mGluR expression might have a pathological connection to PD progression or exaggeration; therefore, glutamate receptors are exciting targets for novel drug design.

Assessment of both PD patients and animal brains have reported upregulation of mGluR5 expression, which is proportionally related to the elevated levels of α-synuclein (αS) aggregation [15], a well-known hallmark of PD. In contrast, some studies have reported that αS selectively binds to mGluR5, not mGluR3, at its N-terminal region and stimulates microglia-mediated neuroinflammation [16]. Small, single-site trials of a highly specific radiopharmaceutical of mGluR5 in PD have been conducted to enlighten pathological connection; however, the outcome is complicated or inconclusive [17,18]. This review discusses the most recent findings on mGluR5 in PD progression, highlighting its importance in designing novel therapeutics and diagnosing PD.

## 2. Localisation of Group I mGluRs in the Brain

The members of the group I mGluRs are widespread throughout the brain. mGluR1 is highly expressed in the cerebellar cortex neurons, olfactory bulb, lateral septum, globus pallidus, entopeduncular nucleus, ventral pallidum, magnocellular preoptic nucleus, and thalamic nuclei [19,20,21]. mGluR5 is mostly expressed in the telencephalon, specifically in the cerebral cortex, hippocampus, subiculum, olfactory bulb, striatum, nucleus accumbens, and lateral septal nucleus [22,23,24]. A high expression of mGluR5 could be seen in the superficial dorsal horn of the spinal cord [8]. In the CA3 region of the hippocampus, cerebellum, olfactory bulb, and thalamus, mGluR1 has been observed to be highly expressed, while mGluR5 has high expression in the CA1 and CA3 region of the hippocampus, cortex, striatum, and olfactory bulb [25]. A comparative study using rat and monkey brains showed that high-dense mGluR1 expression was found at the plasma membrane, whereas a bulk amount of mGluR5 was expressed in the intracellular compartment of the substantia nigra. Plasma membrane-bound group I mGluRs are primarily extrasynaptic or expressed in the main body of symmetric, GABAergic, striatonigral synapses in rats and monkeys [21].

Both receptors have shown subtype-specific variation in their localisation and expression during development of the brain [26,27]. For example, mGluR1 expression increases gradually in both the hippocampus and neocortex during the development phase [26]. In the cortex, mGluR5a expression reaches a peak during the second postnatal week and falls subsequently [26], while mGluR5b mRNA level increases postnatally, and this subtype is predominantly expressed in adults [28].

The activation and expression pattern of group I mGluRs might have a regulatory role in various aspects of neurogenesis and synaptogenesis during the development phase of the cortex [28,29]. A pattern of distribution of group I mGluRs in a region of the brain relates to their distinct functions. Microscopic analysis of mGluR1 and mGluR5 showed that they are localised outside postsynaptic membranes in the perisynaptic annulus around the synaptic junctions [30]. Group I mGluRs are also present in peripheral cells outside the brain, regulating nociceptive signalling and inflammatory pain [31].

In terms of cellular specificity, although most of the mGluRs are expressed in the neuronal cells, exceptionally, mGluR3 and mGluR5 have been evidently expressed in the glial cells throughout the brain. However, cell genotypic variation would be the reason for the difference in expression of mGluRs in different cell types. To clarify this context and establish a database of mGluRs expression intensity in different cell types in the cortex, Zhang et al. (2014) [32] have conducted a high-resolution transcriptome using RNA-Seq of purified neurons, astrocytes, microglia, and various maturation states of oligodendrocytes from mouse cortex. That study indicates that the mGluR1 is mostly expressed in neurons, whereas mGluR5 has more intense expression in the astrocytes than in neurons in the cortex.

## 3. Group I mGluRs Signalling in Brain

### 3.1. Basic Signalling of Group I mGluRs

Both members of group I mGluR contain an extracellular domain for natural ligand binding and a seven-transmembrane domain (7TM) for synthetic allosteric modulator binding. The mGluR1 ligand binding site has a crystal structure that separates two globular domains by a hinge region and expresses the receptors’ resting or active form by opening or closing, respectively, in the absence of ligand [33]. Human mGluR1 and mGluR5′s crystal structures of the isolated 7TM domain have been well studied [34,35]. Interestingly, these structural studies found that the mGluR1 has a large β-hairpin confirmation at the 2nd extracellular loop position, like the class A GPCRs. Another interesting observation was that the transmembrane region of mGluR1 could form a dimer by TM1–TM1 interactions, and these interactions are stabilised by cholesterol molecules [34].

Group I mGluR activation has been reported to induce myriad oscillatory responses of distinct frequencies largely due to a single amino acid residue in the G-protein coupling domain of mGluR1 (D854) and mGluR5 (T840) [25]. Furthermore, the lipid content of the plasma membrane might have an influence on the activity of group I mGluRs. Both members of this group have been seen to be present in membranes with a lipid augmented environment [36,37]. However, not any of these receptors have been seen to be associated with the lipid-rich rafts, suggesting that the association might be transient. A study reported that this association between lipid raft and mGluR1 depends on the cholesterol content of the membrane and could be improved by the agonist binding [38]. The TM5 and the third intracellular loop of the receptor has a cholesterol-binding motif that increases cholesterol levels in the membrane, enhancing the agonist-mediated activation of the receptor. However, depletion in the cholesterol level inhibits the mGluR1-dependent extracellular signalling-regulated kinase (ERK) signalling activation [25,38]. These data indicate association and positive regulation of group I mGluR signalling activation by the lipid rafts and membrane cholesterol.

Group I mGluRs are positively coupled to the G-protein Gαq/11, which at the downstream stimulates phospholipase Cβ1 (PLCβ1) and activate diacylglycerol (DAG) and inositol-1,4,5-triphosphate (IP3). The IP3 receptors (IP3R) then trigger the intracellular Ca^2+^ release [8], whereas DAG at the plasma membrane, together with extracellular Ca^2+^, activates protein kinase C (PKC) and activates phospholipase D (PLD), phospholipase A2 (PLA2), and mitogen-activated protein kinase (MAPKs) pathways [39]. The activation of PKC via mGluR5 can also stimulate NMDAR [40]. However, N-methyl-D-aspartate receptor (NMDAR)-dependent activation of calcineurin, a Ca^2+^ channel-dependent phosphatase, reverse the PKC-mediated desensitisation of mGluR5 [41]. Additionally, mGluR1 can upregulate the NMDAR cascade in cortical neurons through Ca^2+^-, calmodulin-, and Src-dependent proline-rich tyrosine kinase (Pyk2) activation [42]. In addition, mGluR1/5-mediated Homer protein interactions are also significant. Homer can phosphoryl IP3 and activate ryanodine receptors and Shank proteins, which are part of the NMDAR protein complex [43,44]. The coupling of Homer proteins and mGluR1/5 also activates Akt via involving phosphoinositide 3-kinase (PI3K), phosphoinositide-dependent kinase (PDK1), and PI3K enhancer (PIKE), which leads to neuroprotection (Figure 1) [45,46]. Although group I mGluRs bind to Gα_q/11_, overexpression of these receptors showed coupling to Gαs and Gα_i/o_ as well. Similarly, mGluR1a has been shown to couple to Gα_i/o_, leading to cAMP stimulation in overexpressed Chinese hamster ovary (CHO) cells [47]. This example suggests that group I mGluRs could couple to a variety of G-proteins, and understanding them might reveal endogenous receptor mechanisms in native form, which could lead to understanding these receptors mechanisms in vivo as well.

Further, group I mGluRs also modulate the ERK signalling cascade through IP3-stimulated Ca^2+^ release, Homer proteins, and Pyk2 [48,49]. Activation of ERK is important for the modulation of cell growth, differentiation, and survival, as well as the increment of neurotrophic factors such as brain-derived neurotrophic factor (BDNF) [50], indicating group I mGluR-mediated neuroprotection could rely on activation of ERK signalling. However, as discussed above, mGluR5 is more highly expressed in glial cells than in neurons, specifically in the astrocytes (Figure 2), where they form complexes with IP3 and increase intracellular Ca^2+^ to facilitate glutamate release and contribute to the apoptosis of astrocytes [51,52,53,54]. Studies also found that mGluR5 activation in cortical and hippocampal astrocytes can stimulate MAPK pathways and PLD signalling [55,56]. Selective activation of mGluR5 by an agonist inhibits microglial-activation and associated neuroinflammation and neurotoxicity via Gαq-signal transduction pathway [57].

**Figure 1 biomedicines-10-00864-f001:**
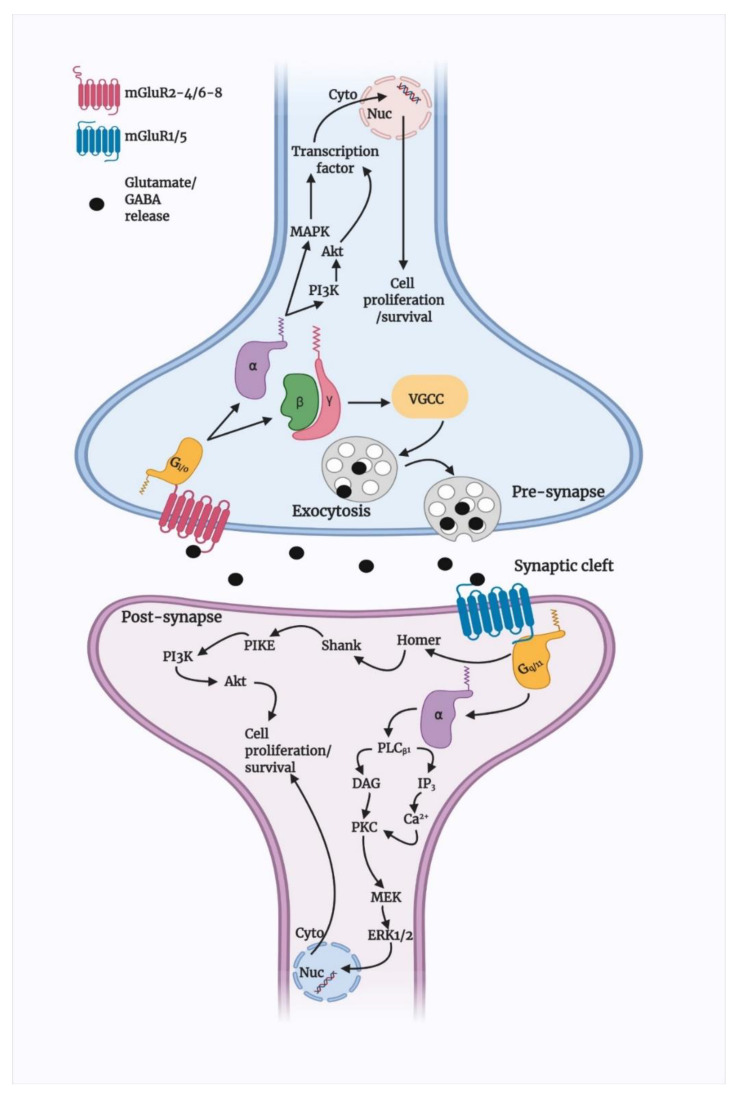
Schematic presentation of mGluRs cell signalling pathways. Likely, other GPCRs and mGluRs are located at the cell membrane that binds to extracellular substances and transmits signals to intracellular molecules called G-protein (G-α, -β, and -γ) Upon agonist activation, both group II and III mGluRs are coupled predominantly to Gi/o proteins, which mediate the downstream inhibition of adenylyl cyclase activity via Gα_i/o_, decreasing the levels of cAMP. Gβγ subunits modulate voltage-dependent ion channels, inhibiting Ca^2+^ and limiting presynaptic glutamate or GABA release. Group II and III mGluRs also activate PI3K/Akt and MAPK pathways and enhance neuroprotection by increasing the production of neurotrophic factors [58]. In the post-synapse, following glutamate binding, group I mGluRs uncouple Gα_q/11_ proteins, stimulate PLCβ1, and activate DAG and IP3, which increase intracellular Ca^2+^ efflux. Both Ca^2+^ and DAG activate PKC, which has been proposed to activate MEK/ERK1/2 signalling. Furthermore, mGluR1/5 interacts with Homer proteins, which activate Shank proteins. This complex of Homer proteins and group I mGluRs activates Akt through a mechanism that involves PI3K, PDK1, and PIKE, promoting neuroprotection [8]. Physical interaction between elements is represented by a continuous line.

### 3.2. Group I mGluR Desensitisation and Trafficking

Many GPCRs undergo desensitisation via activation of the second messenger pathway to protect receptors from prolonged over-stimulation. Desensitisation results from the uncoupling of a specific GPCR from the respective G-protein involved. Several GPCR desensitisation mechanisms have been assessed, and the observations suggest that the process depends on several facts, including the type of receptor, type of ligand, and type of system [59,60,61]. Phosphorylation plays a crucial part in some GPPCR desensitisation; phosphorylation leads the receptor to bind to adapter proteins, such as β-arrestin, that interferes with G-protein coupling and leads to the second messenger pathway generation [59]. For the others, endocytosis plays a crucial part in desensitisation [61].

Several kinase-dependent desensitisations of group I mGluRs have been tested so far, and it has been seen that PKC is important in the agonist-mediated desensitisation of group I mGluRs. For example, phosphorylation of mGluR1a by PKC leads to the desensitisation of the receptor [62]. Interestingly, activation of PKC has been shown to affect the mGluR1 pathway coupled to Gαq, but it does not affect the coupling of the receptor to the cAMP pathway. These data indicate selective desensitisation of mGluR1 via PKC activation [10]. The desensitisation of the mGluR5 has been well studied rather than mGluR1. The presence of several serine/threonine residues in mGluR5 is presumably involved in the PKC-mediated desensitisation process. mGluR5 has a calmodulin-binding site, and in the basal state, calmodulin interacts with mGluR5 at the region of the S881 and S890 amino acid residue sites of the receptor, and PKC has been shown to phosphorylate these two-binding sites [63]. In contrast to PKC-mediated inhibition of calmodulin binding to mGluR5 via phosphorylation, calmodulin can inhibit PKC-dependent phosphorylation of the receptor [64]. These data suggest that PKC-dependent phosphorylation and calmodulin-binding counterbalance each other. PKA, another second messenger-dependent protein kinase, shows the opposite effect on the group I mGluR desensitisation process. PKA activation results in the dissociation of adapter proteins from the C-terminal of the receptor and leads to the inhibition of the receptor endocytosis and agonist-dependent desensitisation of mGluR1 [62]. For many GPCR desensitisations, G-protein coupled receptor kinases (GRKs) plays a crucial role. GRK-mediated phosphorylation of specific residues of the receptor results in the binding of β-arrestin that uncouples the receptor from the respective G-proteins [59,60,61]. It has been suggested by several studies that GRKs could regulate desensitisation of both members of group I mGluR when heterologously expressed in HEK293 cells and primary neurons [65,66,67]. GRK2 has been involved in the desensitisation process of mGluR1 and mGluR5, which seems to be phosphorylation independent [66,68]. Conversely, GRK4 has shown the selective desensitisation of mGluR1 in cerebellar Purkinje neurons but not mGluR5 [67]; likewise, GRK5 affect mGluR1-mediated Purkinje turnover [69]. Since GRKs typically are not limited to their substrate specificity, it has been challenging to find GRK-mediated residual modification in group I mGluRs.

## 4. Group I mGluRs in Parkinson’s Disease

### 4.1. Alterations in Basic Signalling

Group I mGluRs have distinct roles and functions in the brain, which have not been well characterised, specifically in disease pathology such as PD. Positron emission tomography (PET) imaging of a chronic PD rat striatum showed transiently increased expression of mGluR1, but not mGluR5, which dramatically decreased with disease progression [70]. In addition, dynamic changes in mGluR1 during disease progression correlate with striatal dopamine transporter downregulation, indicating a correlation with a pathological decrease in general motor activity. Evidence suggests that mGluR5-mediated neurotransmission increases PD and leads to levodopa (L-DOPA)-induced dyskinesia (LID); LID could result from aberrant dopamine-related neural plasticity at glutamate corticostriatal synapses in stratum [71].

The binding potential of the mGluR5 receptor has been seen decreasing in the bilateral caudate-putamen (CP), ipsilateral motor cortex, and somatosensory cortex, eventually in both PD and LID pathology. However, 6-OHDA-induced PD rats did not show any significant alteration in mGluR5 binding potential in those regions upon L-DOPA treatment [18]. L-DOPA treatment substantially increased mGluR5 uptake at the contralateral motor cortex and somatosensory cortex and has been found positively related with abnormal involuntary movement. However, acute regulation of mGluR5 in the cortical astrocytes causes oscillatory Ca^2+^ and synaptic release of neurotransmitters. Certain changes in Ca^2+^ signalling might bring about interaction between mGluR5 and NMDA receptors; it has been evident in rat hippocampus that mGluR5 enhanced phosphorylation of NR2B [72]. This evidence suggests that negative allosteric modulation of mGluR5 or any of group I mGluR may provide symptomatic alleviation in Parkinson’s disease via reducing overstimulation of basal ganglia nuclei.

### 4.2. Interaction with α-Synuclein

Aggregation of oligomeric αS species, synaptic dysfunction, and subsequent neuronal cell death are the key pathophysiological feature of synucleinopathy, including PD, which is known, but the precise molecular mechanism or the nature of toxicity during aggregation is unclear. Recent PD-related studies concentrate on extracellular soluble αS oligomers because of their critical role in PD pathogenesis and progression. It is now supposed that αS is released and propagates between neurons in a prion-like fashion [73], so different αS species (monomers, multimers, oligomers, and fibrils) obtain access to the extracellular space, and their postsynaptic action impairs neuronal communication and plasticity. Inconsistent with this hypothesis, a recent study showed that PrP^C^ plays the role of cell surface binding associate for β-sheet-rich protein aggregates [16,74,75], precisely soluble oligomeric αS via NMDAR activation, evoked by mGluR5 [73]. Oligomeric species of αS interact with PrP^C^ via mGluR5, activate Src tyrosine kinase family (SFK) kinase and NMDAR2B, and causes synaptic dysfunction [73]. Thus, blockade of mGluR5-mediated NMDAR phosphorylation could rescue them from synucleinopathies by harmonising synaptic and cognitive functions.

Contrarily, selective modulation of mGluR5 degradation and its intracellular signalling showed protection against αS neurotoxicity in microglia. mGluR5, but not mGluR3, selectively binds to αS at the N-terminal region. This interaction promotes lysosomal degradation of mGluR5 and abrogates neuroinflammation mediated by αS. Treatment with mGluR5 agonist CHPG, not antagonist MTEP, rescued them from αS-induced microglial activation and cytokine release by reducing nuclear factor κB (p65) and TNF-α activation [16,76] (Figure 3). Although it is still to be assured that activation or deactivation of mGluR5 in specific brain cells or regions could protect neuronal cytotoxicity, it is confirmed that the mGluR5-αS complex has a crucial role in PD pathology, and dissociation of this complex could modify pathogenicity.

### 4.3. Modulation of Apoptotic Signalling

As we reviewed in the previous section, the signal transduction of group I mGluRs has a diverse role in neurogenesis, neural progenitor cell proliferation, differentiation, and protection. Contrarily, both genetic and pharmacological blocking of group I mGluRs negatively affects cortical, hippocampal and striatal progenitor cell growth and survival [77]. mGluR5 activation also promoted cerebellar granule cell survival [78] and increased release of soluble β-amyloid precursor protein (APP) derivatives from the cortex and hippocampus that protects from neurotoxicity in AD [79].

Although mGluR5 activation has been important for neuronal survival and proliferation, overactivation of glutamatergic transmission promotes dopaminergic neuronal loss. Thus, selective modulation of this excitatory neurotransmitter has been proposed as a promising target in PD. As mGluR5 is widely distributed throughout the basal ganglia, selective antagonism of this receptor by MPEP/MTEP or negative allosteric modulators (NAMs) has been shown to promote neuroprotection in PD [80]. Selective modulation of group I mGluRs was found to inhibit toxicant or environmental stress-induced dopaminergic neuronal loss via modulating PI3K and JNK phosphorylation [81]. Specific modulation of mGluR5 via cystic fibrosis transmembrane conductance-regulator-associated ligand (CAL) could prevent rotenone-induced neuronal apoptosis in PD via AKT and ERK1/2 phosphorylation [80]. These studies suggest that selective modulation, but not blockade, of group I mGluRs could protect from apoptotic cell death in PD. Accordingly, a study confirmed that mGluR5 knockout was shown to inhibit oxygen–glucose deprivation-induced astrocytic deaths but did not protect from necrotic cell death [82]. In response to oxygen–glucose deprivation, mGluR5 stimulates Gα_q/11_ and activates downstream PLC, which interacts with IP3. This interaction results in an increased level of intracellular Ca^2+^ release and cell death. To protect from mGluR5-mediated apoptotic cell death, the blockade of mGluR5 by selective antagonist MPEP/MTEP showed sufficient protection against apoptotic cell death of cortical astrocytes [81]. Up-regulation of mGluR5 selectively regulates apoptosis via PLC and increasing intracellular Ca^2+^. Additionally, targeting mGluR5 activation and mGluR5–Homer interaction could counteract this scenario by modulating intracellular Ca^2+^ release and astrocytic apoptosis.

Primarily, inhibition of caspase activation prevents cellular apoptosis; however, recently it has been seen that inhibition of caspase-dependent apoptosis shifts the programmed cell death pathway towards necrosis [83]. Necrosis is an unregulated form of cell death characterised by cell swelling and disruption [84]. However, the regulated version of necrosis is necroptosis, regulated by a specific stimulus such as RIP1 kinase or MLKL. Although the relation of metabotropic glutamate receptor activation or inhibition and necrosis has not been established yet, activation of mGluR groups II/III showed neuroprotection via inhibition of necroptosis [85]. Both orthostatic agonist and PAM of mGluR II/III modulated caspase-3 activation reduced necrotic nuclei and up-regulated pro-survival ERK1/2 phosphorylation MPP+-treated differentiated SH-SY5Y dopaminergic cells of a PD model. Inhibiting group I and stimulation of group II/III could be beneficial for motor symptom improvement in PD and reduced dopaminergic neurodegeneration in the substantia nigra [86,87]. Thus, it is speculated that group I mGluR plays some role in necroptosis, which has not been evaluated yet.

## 5. Neuroimaging of Group I mGluRs for Diagnosis and Therapeutic Development

The progression of PD significantly fluctuates glutamate receptors expression. Thus, a specific group I mGluR tracer could image the stage-to-stage progression of PD, which could help with therapeutics development. For example, longitudinal positron emission tomography (PET) imaging using [11C]ITDM (N-[4-[6-(isopropylamino) pyrimidin-4-yl]-1,3-thiazol-2-yl]-N-methyl-4-[11C]methylbenzamide) and (E)-[11C]ABP688 [3-(6-methylpyridin-2-ylethynyl)-cyclohex-2-enone-(E)-0-[11C] methyloxime] ligands for mGluR1 and mGluR5, respectively, showed dramatic changes in striatal non-displaceable binding potential (BPND) values of both receptors with PD progression [70]. Analysing striatal BPND values for both receptors revealed that mGluR1, not mGluR5, increases temporarily at the early onset of PD symptoms and declines with the pathological progression of the disease. Furthermore, this decrease of striatal mGluR1 is associated with impaired general motor activities. However, another study [12] used both DAT imaging agent [11C]PE2i and mGluR5 antagonist [18F]FPEB. They reported that DAT tracers are better-diagnosing tools for PD, while together with mGluR5 tracers, regional neurotransmitter abnormality could be explained, while probing with radioligands such as [11C]ABP688 or [18F]FPEB could measure availability and interaction with mGluR5 [88]. Although these studies showed the potential application of glutamatergic tracers in PD diagnosis, measuring true biological differences using them is yet a concern. For example, [18F]FPEB showed dilute dorsal striatum of mGluR5 but concentrated in the ventral striatum in reality, which is vice-versa. It warrants further exploration of mGluR5 tracers to convert them into potential biomarkers.

Neuroimaging could identify specific preferential binding sites that could reveal a new pharmacological target. Autoradiography study using radioligand [3H]AZD9272 revealed fenobam, selective mGluR5 antagonist, ventral striato–pallido–thalamic circuit binding potential [89]. Like AZD9272, fenobam also showed psychosis-like phenomena during clinical trials [85], associated with both compounds binding to different brain regions. This could potentially help in understanding the pathophysiology of psychotic disorders like schizophrenia and identify novel antipsychotic treatment.

Moreover, mGluR5 tracers are diagnostic tools that could reveal their association with other motor dysfunction diseases such as LID. PET imaging with [18F]FPEB showed rapid mGluR5 uptake in the caudate–putamen region after levodopa treatment that caused abnormal involuntary movement [18].

Although many mGluR5 receptor antagonists have been successfully used to label mGluR5 in vitro, the PET tracers have failed in vivo to meet expectations. The failure was due to high nonspecific binding, unfavourable brain uptake kinetics, and/or limited metabolic stability. For example, [18F]FPEB showed binding potential weaker than DAT tracer [11C]PE2i during PD patient brain diagnosis [12]. It is through ushering that several mGluR5 PET tracers were made for clinical trial, namely [18F]FPEB, [18F]FPEP, [18F]SP203, [11C]MPEP, and [11C]ABP688. Yet, a few factors limit the widespread use of imaging agents of the human brain, for example, the short physical half-life of carbon-11 (t1/2 = 20 min). Among other factors, CNS PET ligands mostly depend on brain kinetics and in vivo metabolism, so the probability of success depends on these criteria improvement. So far, mGluR5 radioligands have shown their high utility in disease pathology characterisation and drug development programs. Undoubtedly, mGluR5 PET ligands are emerging targets to uncover several psychiatric and neurological diseases questions where mGluR5 is hypothetically involved.

## 6. Emerging Therapeutics and Prospective Targets of Group I mGluRs in PD Therapy

As mGluR1 and mGluR5 are widely expressed in the basal ganglia structures, especially at postsynaptic sites [90], and a high expression of mGluR1 receptors can be found in the globus pallidum (GP), substantia nigra pars reticulata (SNr), and striatum, therefore they might be involved in PD pathogenesis. A study showed that antagonism of mGluR1 using negative allosteric modulators (NAMs) did not reduce LID in PD; only blocking of mGluR5 showed a promising reduction of dyskinesia [91]. Some studies also showed that using the mGluR5 NAMs. such as 2-methyl-6-(phenylethynyl)-pyridine (MPEP), mavoglurant, dipraglurant, fenobam, and 3-((2-Methyl-4-thiazolyl)ethynyl)pyridine (MTEP), were shown to ameliorate motor deficits in PD animals [8,91,92]. Fenobam and AZD9272 have been reported to induce psychosis-like adverse events. Varnas et al. (2020) reported from a PET study of the human brain that both antagonists bind to monoamine oxidase-B (MAO-B), which reveals a new understanding of psychosis-like adverse effects and could generate new models for the pathophysiology of psychosis [93]. MPEP treatment significantly ameliorated akinesia in 6-hydroxydopamine (6-OHDA)-induced rodents and decreased LID in MPTP treated monkeys [94,95]. Chronic treatment with MPEP in MPTP-treated PD monkeys for 1 month was found to inhibit LID [96]. Administration of MTEP also showed a significant decrease of dyskinesia in MPTP-treated monkeys [95] and 6-OHDA-lesioned rats [97]. Several studies with different other NAMs such as mavoglurant [98], dipraglurant [99], and fenobam [100] also reported similar anti-parkinsonism and a decrease in LID of L-DOPA in different PD models. MPEP chronic treatment was shown to attenuate DA neuronal loss and prevented microglial activation in SNpc of 6-OHDA treated or MPTP-treated rats [101,102]. Moreover, MTEP local infusion in the striatum was reported to attenuate 6-OHDA-induced activation of ERK1/2 signalling that is associated with dyskinesia [103]. Different antagonists of mGluR5, including AFQ056 (mavoglurant) and ADX-48621 (dipraglurant), are currently being tested in humans as anti-dyskinetic drugs. These drugs are well tolerated and have still not been reported to worsen PD motor symptoms [104], which is encouraging and supportive to study further and develop mGluR5-related compounds as potential neuroprotective drugs in PD.

### 6.1. Regulation of Autophagy

Group I mGluRs are potential regulators of several autophagic signal transducers that contribute to the pathophysiology of neurodegenerative diseases such as AD and HD. However, the role of members of this mGluR group has not been evaluated yet in the PD model; hence, in this section, we prospected a few potential targets that might interest mGluR-mediated autophagy-based therapeutic development. Optineurin is a multifunctional cellular network processor protein that regulates membrane trafficking, inflammatory response, and autophagy, and mGluR5-mediated autophagic signalling is regulated by optineurin [105]. mGluR5 couples to canonical Gαq/11 and activates autophagic machinery via mTOR/ULK1 and GSK3β/ZBTB16 pathways. In this process, mGluR5 promotes intracellular Ca^2+^-influx and signals ERK1/2; interestingly, optineurin regulates mGluR5-mediated Ca^2+^ and mTOR/ULK1 and GSK3β/ZBTB16 pathway activation. Although crosstalk between optineurin and mGluR5 and their contribution to neurodegenerative diseases pathology is now known [105], downstream signalling remains largely unknown.

In addition, long-term use of mGluR5 NAM (CTEP) attenuated caspase-3 activation, neuronal loss, and apoptosis in both heterozygous and homozygous knock-in HD mice models [106], which occurred via activation of GSK3β/ZBTB16-mediated autophagy. Inhibition of mGluR5 attenuates autophagosome biogenesis-related kinase ULK1 and increases autophagy factor ATG13 and Beclin1. In addition, inhibition of mGluR5 by CTEP reduces aberrant phosphorylation of the PI3K/Akt/mTOR signalling cascade [107] that promotes ULK1 activity and autophagy. Antagonism of mGluR5 via selective NAM (CTEP) promotes aggregated protein clearance by autophagy activation and facilitates CREB-mediated BDNF expression in the brain, fostering neuronal survival and reducing apoptosis. In addition, chronic use of CTEP for 24 weeks was shown to reduce the Aβ burden in APPswe/PS1ΔE9 mouse hippocampus and cortex; however, CTEP at 36 weeks became ineffective [108]. Reflecting that mutation at APP in the advanced disease stage could alter mGluR5 expression and mGluR5-mediated ZBTB16 and mTOR signalling in the brain. Inhibition of mGluR5 and subsequent mTOR phosphorylation could also alleviate inflammatory responses by decreasing IL-1β expression that might have been correlated with the activation of autophagy [109].

### 6.2. Gut-Brain Axis

Braak et al. (2003) [110] postulated that αS pathology could spread from gut to brain, and that the vagus nerve plays an essential role in this process. A study showed that αS injection at the duodenum and pyloric muscularis layer led to αS accumulation at the dorsal motor nucleus and later spread in caudal portions of the hindbrain, locus coeruleus, basolateral amygdala, and SNpc [111]. mGluR5 antagonism (by MPEP) significantly affects peripheral afferent ending gastric vagal circuitry [111]. Thus, suppressing primary sensory endings via mGluR5 antagonist could rescue them from αS-pathy and associated neurodegeneration and behavioural deficits.

## 7. Conclusions

Fundamental research into mGluR neurobiology has directed the identification of several lead compounds for treating NDs. Over the past decades, advanced research has been conducted to identify selective allosteric ligands and modulators of mGluRs, unveiling the prevalence and capacity for biased agonism and modulation of this receptor. However, the role of mGluR signalling pathways that can differentiate therapeutic and adverse effects needs further investigation. A better understanding of the biased, canonical, or non-canonical signalling of mGluRs might facilitate rational drug design that would preferentially modulate pathways associated with positive therapeutic outcomes while avoiding off-target adverse effects.

PD pathogenesis decreases dopaminergic neurotransmission, while at the basal ganglia glutamatergic signalling increases dopamine release in the SNpc region as a compensatory mechanism. However, excessive glutamate release by the hyperactivation of glutamate receptors could be pathogenic for the PD brain. Excessive activation of NMDARs led to increased Ca^2+^ influx and increased production of ROS, aggravating PD pathogenesis. Considering the treatment strategy for PD, until now, dopamine replacement is the gold standard, although the success rate is not ideal. Finding an alternative target could compensate for this therapeutic gap, for example, Nedd4-2 knockdown attenuated astrogliosis and reactive microgliosis by reducing glutamate excitotoxicity. NMDAR antagonists or mGluR5 NAMs have been shown to attenuate motor symptoms in the PD model. Thus, further in-depth research into mGluRs signalling, subsequent activation, glutamate release, and related regulation of the central neurotransmission could decipher the molecular mechanism of PD pathogenesis. It could also provide an effective therapeutic target(s) to intervene in PD.

## Figures and Tables

**Figure 2 biomedicines-10-00864-f002:**
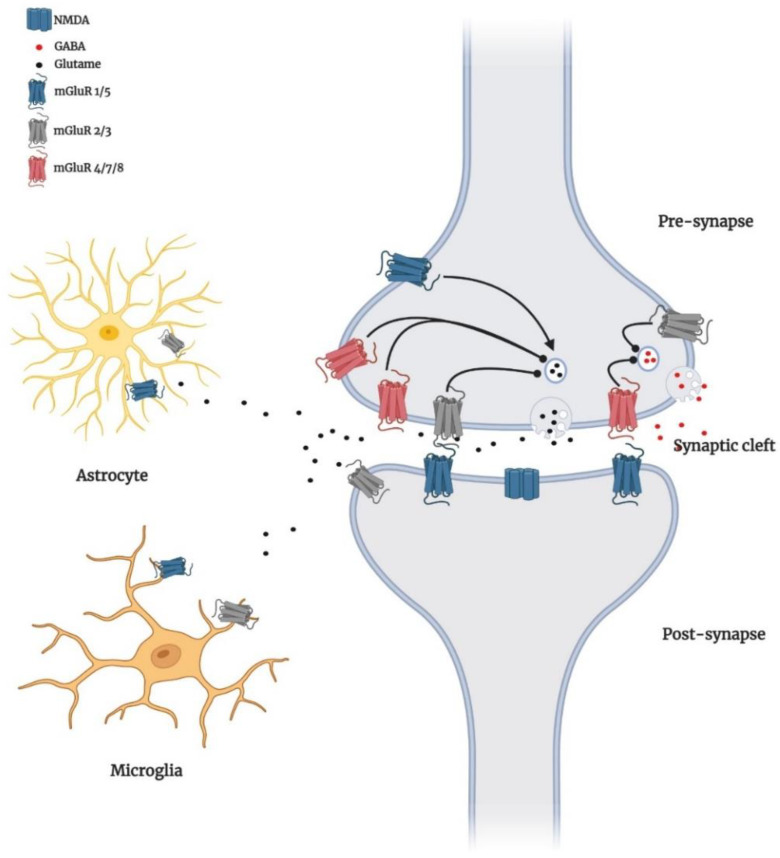
Schematic display of the distribution of mGluRs. Group I mGluRs (blue) are localised post-synaptically, and group II (gray) and III (red) receptors are localised in presynaptic neurons. However, exceptions occur; for example, mGluR5 and mGluR3 are widely expressed in glial cells (astrocyte and microglial cells) [54], although signalling and consequences of mGluR in glial cells have yet to be fully uncovered but are now considered as an emerging key site of synaptic mGluR regulation. In presynaptic locations, mGluRs 2, 3, 4, 7, and 8 are generally found in extrasynaptic areas and inhibit the release of glutamate (black circles) or GABA (red circles). In contrast, group I receptors promote release when present. At the postsynaptic terminal, the glutamate-gated ion channel NMDA responds to glutamate with increases in intracellular sodium or calcium, promoting cell excitability. mGluR5 and NMDA receptors are closely linked signalling partners reciprocally regulated by phosphorylation. Postsynaptic mGluR2/3 receptors couple to cAMP inhibition.

**Figure 3 biomedicines-10-00864-f003:**
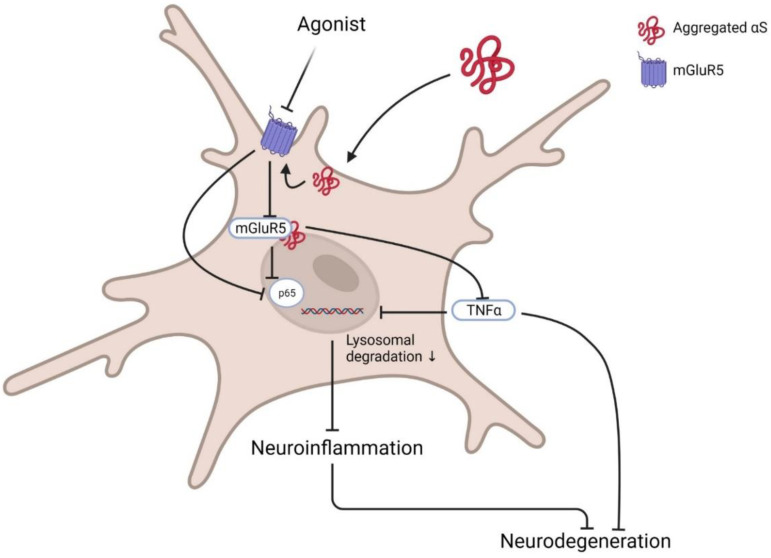
Basic pathway of mGluR5 agonist in PD. Post activation of microglia by the presence of αS at the membrane inactivates membrane receptor mGluR5 and its downstream G-protein. Sequentially activated transcription factors NFκB p65 and TNFα enhance inflammatory cytokine release and induce neurodegeneration. Agonists such as CPHG could ameliorate this pathway by increasing mGluR5 expression via reducing lysosomal degradation, further downregulating related inflammatory signalling activation and subsequent inhibition of microglia-mediated inflammation to prevent neurotoxicity. Created with BioRender.com (accessed on 31 March 2022).

## Data Availability

Not applicable.

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
