# Peer review of "Group I mGluRs in Therapy and Diagnosis of Parkinson’s Disease: Focus on mGluR5 Subtype"

_biomedicines, 2022, doi:10.3390/biomedicines10040864_

Round 1

Reviewer 1 Report

This a review article about an interesting topic. Its structure is generally good.  Despite that, there are several things that need to be improved before publication.

Regarding the introduction section, it could be more detailed and explain the background of the relation between mGluRs and Parkinson’s disease in bigger depth.

English language and style are fine, but several corrections should be made before publication:

In the title: “Parkinson’s diseases” should be replaced by “Parkinson’s Disease”.

The term “signalling” should be replaced by “signaling” in lines 16, 25, 40, 51, 97, 108, 109, 132, 133, 157, 166, 171, 180, 193, 237, 256, 277, 286, 434 ,440, 447, 454, 471 and 473.

In line 107, the second period should be erased.

In line 322 the phrase “has yet not been established” should be rephrased as “has not been established yet”.

Finally, the conclusions could perhaps be rewritten in a more detailed and critical way.

Author Response

This a review article about an interesting topic. Its structure is generally good.  Despite that, there are several things that need to be improved before publication.

Regarding the introduction section, it could be more detailed and explain the background of the relation between mGluRs and Parkinson’s disease in bigger depth.

Response: Thank you for this suggestion. We have updated our introduction section. We have added more details how glutamate release affect or exacerbate PD by promoting neuronal cell death (line 53-58).

English language and style are fine, but several corrections should be made before publication:

In the title: “Parkinson’s diseases” should be replaced by “Parkinson’s Disease”.

Response: Thank you this suggestion, we have corrected according to suggestion.

The term “signalling” should be replaced by “signaling” in lines 16, 25, 40, 51, 97, 108, 109, 132, 133, 157, 166, 171, 180, 193, 237, 256, 277, 286, 434 ,440, 447, 454, 471 and 473.

Response: Thank you for this suggestion. We have followed British spelling format throughout our manuscript. Therefore, if we change “signalling” into “signaling”, we have to change all other spelling as well.

In line 107, the second period should be erased.

Response: Corrected in revised version.

In line 322 the phrase “has yet not been established” should be rephrased as “has not been established yet”.

Response: Corrected the phrase as suggested.

Finally, the conclusions could perhaps be rewritten in a more detailed and critical way.

Response: Appreciate this suggestion, we have now added another paragraph in that section, which is detailing and focusing our reviewed topics.

Reviewer 2 Report

The review by Shofiul Azam and co-authors is an up-to-date and provides an insight into metabotropic glutamate receptors (focus on mGluR5 subtype) in diagnosis and possible therapy of the Parkinson's disease. Overall, the article is well written though some corrections required to present the data in the best value:

Point 1: Fix Parkinson's diseases to Parkinson's disease (lines 2-3)

Point 2: A lot of abbreviation partially unexplained or explained later in the text, or with inconsistency in presenting (e.g., InsP3 vs IP3) make it poorly understandable. Please present a separate abbreviation list.

Point 3: Please check the super and subscript writing, which is lacking in the entire text (e.g. PrPc; Ca2+; [18F]FPEB etc.).

Point 4: In lines 269-272, the authors extrapolate the results obtained for the amyloid-β peptide (Ref 75) to support the alfa synuclein hypothesis. It cannot be directly done, since amyloids behave differently.

Point 5: There is an incohesion in the pathway description across the text, such as 160-161 and 307-310, where the same order of actions according to the authors lead to different outcomes.

Point 6: Fix developmen to development (line 330).

Point 7: Fix glutametargic to glutamatergic (line 378).

Point 8: G-proteins are presented incorrectly on the Figure 1. For the activation they have to be on a membrane but not a cytosol.

Point 9: Please consider add some additional explanation to the left part of the Figure 2 (astrocyte and microglia) or omit it.

Point 10: Please decipher in the description all the brain structures you mention on the Figure 3.

Point 11:  There is no direct link to Figure 4 in the text. I also do not see any words regarding p65 and TNF in the text just in the Fig 4 description. This picture is a redraw from Zhang et al. J Neuroinflammation 2021, 18, 23 (reference 76) and has a mistake as the lysosome is not defined.

Point 12: I would also suggest the authors to expand the outcome of their literature research. They present a lot of interesting information and the conclusion can be formulated more scientifically valuable.

Author Response

The review by Shofiul Azam and co-authors is an up-to-date and provides an insight into metabotropic glutamate receptors (focus on mGluR5 subtype) in diagnosis and possible therapy of the Parkinson's disease. Overall, the article is well written though some corrections required to present the data in the best value:

Point 1: Fix Parkinson's diseases to Parkinson's disease (lines 2-3)

Response: Thank you for this suggestion, we have corrected as indicated.

Point 2: A lot of abbreviation partially unexplained or explained later in the text, or with inconsistency in presenting (e.g., InsP3 vs IP3) make it poorly understandable. Please present a separate abbreviation list.

Response: Appreciate this suggestion. We have corrected in text abbreviations and added abbreviation section after conclusion section.

Point 3: Please check the super and subscript writing, which is lacking in the entire text (e.g. PrPc; Ca2+; [18F]FPEB etc.).

Response: Sincere apology for this typo, we have corrected in our revised version.

Point 4: In lines 269-272, the authors extrapolate the results obtained for the amyloid-β peptide (Ref 75) to support the alfa synuclein hypothesis. It cannot be directly done, since amyloids behave differently.

Response: Agree with reviewer that, although, both Aβ and α-synuclein forms aggregation and behave like prion protein, they are not same. In line 269-271, we have described PrPC is playing role β-sheet-rich proteins conformational change. Later, (line 271-272) we used Ref. 73 to directly indicate NMDAR activation mediating conformational change in α-synuclein. Indeed, ref. 75 used “Aβ-peptide”, but we brought their article provide enough support to our main topic (ref. 73) that PrPC has potential role in β-sheet rich aggregation, since both peptides form β-sheet rich oligomers.

Point 5: There is an incohesion in the pathway description across the text, such as 160-161 and 307-310, where the same order of actions according to the authors lead to different outcomes.

Response: Appreciate this concern, we have modified both sentences for better understanding.

Point 6: Fix developmen to development (line 330).

Response: Thank you, we have corrected the typo.

Point 7: Fix glutametargic to glutamatergic (line 378).

Response: Thank you, we have corrected the typo.

Point 8: G-proteins are presented incorrectly on the Figure 1. For the activation they have to be on a membrane but not a cytosol.

Response: Agree with reviewer that the G-protein should be on membrane. In figure 1 mGlu receptors and their respective G-proteins are located at the pre- and post-synaptic membranes. As there are variety in downstream G-protein activation and respective signalling, we used arrows to specify. Due to this, they are seemingly not in the intracellular membrane. To avoid the cytoplasmic and membrane confusion, we have now labelled “cyto” cytoplasmic nearby “nuc” nucleus. Also, we have defined their location in the figure legend.

Point 9: Please consider add some additional explanation to the left part of the Figure 2 (astrocyte and microglia) or omit it.

Response: Appreciate this suggestion, however, figure 2 representing localisation of different mGluRs. We showed not all but mGluR5 and mGluR3 are expressed in glial cells. Although glial cells are expressing mGluRs but their signalling pathway and subsequent action is yet to uncover completely. We have mentioned few details in figure legend and in text, in our revised version.

Point 10: Please decipher in the description all the brain structures you mention on the Figure 3.

Response: In figure 3, we were showing group I mGluRs dense expression regions in the brain and a possible pathway that selective or non-selective tracers could follow. That figure showing a classical networking system followed by glutamate release and signalling. We proposed to use this networking system as diagnostic tools to predict PD stages and glutamate excitotoxicity in brain. However, we have removed that figure in our revised since it is complex to explain a pathway while each tracer are following similar path and sometime giving non-specific signal.

Point 11:  There is no direct link to Figure 4 in the text. I also do not see any words regarding p65 and TNF in the text just in the Fig 4 description. This picture is a redraw from Zhang et al. J Neuroinflammation 2021, 18, 23 (reference 76) and has a mistake as the lysosome is not defined.

Response: Appreciate this observation. We have corrected our figure by adding lysosomal degradation. We have cited figure 3 in text and mentioned p65 and TNF, as well.

Point 12: I would also suggest the authors to expand the outcome of their literature research. They present a lot of interesting information and the conclusion can be formulated more scientifically valuable.

Response: Thank you for this suggestion, we have added additional paragraph in the conclusion section. We have modified conclusion section by highlighting our topic and what we have discussed in other sections.

Round 2

Reviewer 2 Report

The authors have addressed my comments and concerns, and improved the quality of the manuscript

Author Response

We appreciate the reviewer for their valuable feedback on our manuscript. Thank you